# Linear and Non-Linear Optimal Control Methods to Determine the Best Chemotherapy Schedule for Most Effectively Inhibiting Tumor Growth

**DOI:** 10.3390/biomedicines13020315

**Published:** 2025-01-28

**Authors:** Sotirios G. Liliopoulos, George S. Stavrakakis, Konstantinos S. Dimas

**Affiliations:** 1School of Electrical and Computer Engineering, Technical University of Crete, 731 00 Chania, Greece; sliliopoulos@tuc.gr; 2Department of Pharmacology, Faculty of Medicine, University of Thessaly, 415 00 Larissa, Greece

**Keywords:** chemotherapy, tumor growth inhibition (TGI) mathematical modeling, linear autoregressive with exogenous inputs (ARX), linear quadratic regulator (LQR), state-depended Riccati equation (SDRE), computationally optimized patient-centered chemotherapy, side effects minimization, pancreatic ductal adenocarcinoma (PDAC)

## Abstract

**Background/Objectives:** Cancer is a dynamic and complex disease that remains largely untreated despite major advances in oncology and treatment. In this context, we aimed here to investigate optimal control techniques in the management of tumor growth inhibition, with a particular focus on cancer chemotherapy treatment strategies. **Methods:** Using both linear autoregressive with exogenous inputs (ARX) and advanced non-linear tumor growth inhibition (TGI) modeling approaches, we investigated various single-agent treatment protocols, including continuous, periodic, and intermittent chemotherapy schedules. By integrating advanced mathematical modeling with optimal control theory and methods, namely the Linear Quadratic Regulator (LQR) and the “pseudo-linear” state-space equivalent representation and suboptimal control of a non-linear dynamic system known as the State-Dependent Riccati Equation (SDRE) approach, this work explores and evaluates successfully, more effective chemotherapy treatment strategies at the computer simulation level, using real preclinical data which increases the expectation to be applied in the clinical practice of oncology. **Results:** The integration of these methods provides insights into how different drug administration schedules may affect tumor response at the preclinical level. This work uses mathematical modeling to evaluate the efficacy of various periodic and intermittent chemotherapy treatment strategies, with a focus on optimizing drug doses while minimizing the potential side effects of chemotherapy due to the administration of less effective chemotherapeutic doses. **Conclusions:** The treatment scenarios tested in this study could effectively stop tumor growth or even lead to tumor regression to a negligible or near-zero size. This approach highlights the importance of computational tools for more effective treatment strategies in chemotherapy and offers a promising direction for future research and more efficient clinical applications in oncology as part of a more individualized approach.

## 1. Introduction

Although the number of cancer treatments has significantly increased during the last decades, cancer continues to be a major health threat. Treatment protocols are established through extensive and expensive clinical trials. These trials initially identify the maximum tolerable dose (MTD) and subsequently evaluate its expected effectiveness for an average patient. However, this method, as it operates within the clinical trial framework, does not facilitate a comprehensive assessment of all potential dosing schedules. Thus, the optimal scheduling for systemic treatments, such as chemotherapy, remains largely unexplored.

Within tumor growth modeling, optimal control theory (OCT) could be used to identify the optimal treatment strategy for individual patients. This involves discovering the most effective immunotherapy and/or chemotherapy drug dose levels and treatment schedules while considering the predicted growth of the tumor and the potential risks and benefits of the different treatment options [1]. After developing a tumor growth model based on patient-specific data, optimal control theory can be employed to identify computationally the optimal treatment strategy. This strategy aims to achieve tumor eradication while minimizing potential risks or side effects associated with the treatment, firstly at a computer simulation level. Some of the first applications of optimal control theory to mathematical models of cancer biology and tumor treatment date back to the 1970s [2,3]. The work of Swan and Vincent [3] was the first to apply optimal control in human cancer (IgG multiple myeloma). Following [3], numerous studies on optimal control for mathematical models of cancer therapies, such as chemotherapy and/or immunotherapy, have been published. A small pool of such works, including a plethora of linear and non-linear mathematical tumor growth models, as well as different control methods such as the direct collocation (DirCol), as well as those applied in the present work, i.e., the equivalent “pseudo-linear” representation of the non-linear state-space model with the corresponding *state-dependent Riccati equation* (SDRE) and the linear quadratic control with the linear quadratic regulator (LQR), can be found in [1,2,3,4,5,6,7,8,9,10,11,12,13,14,15,16,17,18,19,20,21,22,23,24,25,26,27,28,29,30,31].

A tumor undergoing treatment, such as chemotherapy, can be conceptualized as a control system based on a set of linear or non-linear state-space differential equations [2,3,4,5,6,7,8,9,10,11,12,13,14,24,32,33]. In this state-space model, the state of the system is given by the population of cancer cells or the tumor mass at a time *t*. The control signal at that time t, is denoted as *u(t)*. Typically, u(t) represents the amount of the administered drug or its impact on healthy tissue and cancer cells. Since chemotherapy can affect both normal and malignant cells, the goal of the control problem is to minimize the number of cancer cells and at the same time maintain a safe level of toxicity for normal tissue. Optimal control theory emerges as a valuable tool for optimizing treatment decisions in the context of tumor growth modeling, helping to improve patient outcomes and increase survival rates. Traditional cytotoxic chemotherapy regimens, often guided by the MTD schedule, can lead to substantial host toxicity and create opportunities for tumor vasculature to regrow and for drug-resistant cell populations to develop during prolonged drug-free recovery periods. Optimal control theory, on the other hand, could be used to design more effective treatment protocols than the standard periodic protocols currently in use. By treating tumor growth inhibition as an optimal control problem, one can model mathematically and study more precisely the dynamics of a tumor’s growth with the goal of minimizing its size by a specific end-time. This approach also seeks to optimize treatment administration, like adjusting the anticancer agent’s doses to the minimum necessary level to minimize the side effects. Such a strategy is crucial for reducing high toxicity levels and minimizing adverse effects associated with the treatment [13,14,15,16,17,18,19,20,21,22,23,24,25,30,31].

In this study, two tumor growth inhibition (TGI) models are presented: (i) a simplified linear autoregressive model with exogenous inputs (ARX) that directly relates tumor growth inhibition and tumor weight at each time point to the anticancer drug dose, in which the PK-PD of the drug is not “explicitly” but “implicitly” taken into account, i.e., in the identified parameters of the linear model, numerical values obtained by using the data of chemotherapy dose (input) versus tumor weight (output) data from preclinical real experiments that were performed previously in our laboratories using human-to-mouse PDAC xenografts [34]; (ii) the well-known non-linear mathematical model of Simeoni et al. [32,33] for tumor growth inhibition, which explicitly takes into account the PK-PD of the drug and whose parameters are determined in a similar way to (i), are combined with optimal control methods, namely the LQR and the SDRE, respectively [35,36,37,38]. Optimal doses of single chemotherapy drugs are investigated by computer simulation across a range of periodic and intermittent treatment schedules. These optimal dosages are then presented, thoroughly evaluated, and discussed.

## 2. Materials and Methods

### 2.1. LQR Optimal Control Background

The *linear quadratic regulator* (LQR) is a commonly used linear optimal control method that provides optimal state-feedback laws for linear systems. It allows for the design of closed-loop stability and high-performance optimal control of linear systems. Consider a linear time-invariant (LTI) discrete-time system described by the following linear equations as:(1)$$\underline{x}\left[k+1\right]=A\underline{x}\left[k\right]+B\underline{u}\left[k\right]\;y\left[k\right]=C\underline{x}\left[k\right],$$
where $A\in R^{n\times n}$, $B\in R^{n\times m}$, $C\in R^{l\times n}$, $\underline{x}\left[k\right]\in R^{n}$ is the state vector, $\underline{u}\left[n\right]\in R^{m}$ is the system’s input (e.g., the chemotherapy dose in the present study), and $y\left[k\right]\in R^{l}$ is the output at each discrete-time instant (point) k, e.g., the tumor’s weight, given that in the preset paper’s cases the dimension l of $R^{l\times n\;}equals$ 1.

LQR seeks to find a control input that minimizes a performance index $J_{LQR}$, typically represented as a quadratic cost function, that weights both the control input u and the system’s states $\underline{x}$. Mathematically, it is expressed as, [35,36]:(2)$${min\;J}_{LQR}=\min\;\{\sum_{k=0}^{\infty }\left({\underline{x}}^{T}\left[k\right]Q\underline{x}\left[k\right]+{\underline{u}}^{T}\left[k\right]R\underline{u}\left[k\right]\right)\},$$
where $Q\in R^{n\times n}$ and $R\in R^{n\times m}$ are real positive semi-definite and positive definite weighting matrices for each state $\underline{x}$ and the control variable u, respectively. The LQR cost function $J_{LQR}\;(2)$ is minimized using the state-feedback controller described by the equation below:(3)$$\underline{u}\left[k\right]=-R^{-1}B^{T}S\underline{x}\left[k\right]≜-K\underline{x}\left[k\right],$$
where $K=R^{-1}B^{T}S$ is the optimal feedback gain and S is a positive definite symmetric matrix and the unique solution of the matrix algebraic Riccati equation (MARE):(4)$$A^{T}S+SA-SBR^{-1}B^{T}S+Q=0,$$

The controllability and observability of $\left(A,B\right)$ and $\left(A,C\right)$, imply the uniqueness of the MARE solution [36].

### 2.2. SDRE Non-Linear Optimal Control Background

The *state-dependent Riccati equation* (SDRE), first introduced by Pearson during the 1960s [37], has become a powerful tool for solving optimal control problems in non-linear systems. By transforming a non-linear state-space mathematical model into a pseudo-linear formulation, referred also as its extended linear form, the SDRE method treats the transformed non-linear state-space system as a sequence of linear time-invariant LTI mathematical models. For each LTI model derived at every time step, a suboptimal solution is computed by solving the corresponding linear matrix algebraic Riccati equation (MARE) [38].

A continuous-time, non-linear state-space mathematical model can generally be represented by the following equation:(5)$$\dot{\underline{x}}=f\left(\underline{x}\left(t\right)\right)+G\left(\underline{x}\left(t\right)\right)\underline{u}\left(t\right),\;\;\underline{x}\left(0\right)=\underline{x_{0}},$$
where $\underline{x}\in R^{n}$ is the state vector and $\underline{u}\in R^{m}$ is the input vector. In many cases, the above non-linear equation can be written in the pseudo-linear form:(6)$$\dot{\underline{x}}=A\left(\underline{x}\right)\underline{x}+B\left(\underline{x}\right)\underline{u},$$
where $f\left(\underline{x}\right)=A\left(\underline{x}\right)\underline{x}$ and $G\left(\underline{x}\right)=B\left(\underline{x}\right)$, with $A\left(\underline{x}\right){\in R}^{nxn}$ and $B\left(\underline{x}\right){\in R}^{nxm}$ where $B\left(\underline{x}\right)\neq 0\forall \underline{x}$. $A\left(\underline{x}\right)$ and $B\left(\underline{x}\right)$ matrices are called state-dependent coefficient (SDC) matrices, and the (6) is then said to be represented in SDC form. If the original system (5) has an equilibrium point at the origin, then $f\left(\underline{x}\left(t\right)\right)$ can be rewritten in a pseudo-linear form as $A\left(\underline{x}\right)\underline{x}$ and then the LQR method could be used. When constructing SDC matrices, it is important to note that the choices for parameterization are not unique. However, the selected parameterization must ensure pointwise controllability for all states $\underline{x}$. This condition can be fulfilled if the state-dependent controllability matrix $M_{c}$ has full rank throughout the time segment during which the SDRE control is being applied, i.e., [38]:(7)$$M_{c}=\left[B\left(\underline{x}\right)\;\;\;A\left(\underline{x}\right)B\left(\underline{x}\right)\;\;\cdots \;\;A^{n-2}\left(\underline{x}\right)B\left(\underline{x}\right)A^{n-1}\left(\underline{x}\right)B\left(\underline{x}\right)\right],$$

Using continuous-time LQR for the time-varying pseudo-linear state-space system (6), the SDRE approach then attempts to determine the sub-optimal controller for the state-space model (6), which minimizes a cost function $J_{SDRE}$, thus driving all states $\underline{x}\left(t\right)$ to zero, i.e.:(8)$${min\;J}_{SDRE}=\min\left\{\frac{1}{2}\int_{0}^{\infty }\left({\underline{x}}^{T}Q\left(\underline{x}\right)\underline{x}+{\underline{u}}^{T}R\left(\underline{x}\right)\underline{u}\right)dt\right\}\;,$$
where $Q\left(\underline{x}\right){\in R}^{nxn}$ and $R\left(\underline{x}\right){\in R}^{nxm}$ are state-dependent matrices that determine the weight for each state and the control input. Thus $Q\left(\underline{x}\right)\geq 0$ and $R\left(\underline{x}\right)\geq 0$ for $\forall \underline{x}$ [38].

Given that the control applied is unbounded, the SDRE cost function $J_{SDRE}\;(8)$ is minimized when:(9)$$\underline{u}\left(\underline{x}\right)=-R^{-1}\left(\underline{x}\right)B^{T}\left(\underline{x}\right)P\left(\underline{x}\right)\underline{x}≜-K\left(\underline{x}\right)\underline{x},$$
where(10)$$K\left(\underline{x}\right)=R^{-1}\left(\underline{x}\right)B^{T}\left(\underline{x}\right)P\left(\underline{x}\right),$$
is referred to as the feedback gain matrix. $P\left(\underline{x}\right){\in R}^{nxn}$ is a symmetric, positive definite matrix and the unique solution of the algebraic SDRE:(11)$$A^{T}\left(\underline{x}\right)P\left(\underline{x}\right)+P\left(\underline{x}\right)A\left(\underline{x}\right)-P\left(\underline{x}\right)B\left(\underline{x}\right)R^{-1}\left(\underline{x}\right)B^{T}\left(\underline{x}\right)P\left(\underline{x}\right)+Q\left(\underline{x}\right)=0\;,$$

The dynamics of the *pseudo-linearized closed-loop* state mathematical model (6) are now described mathematically by the following equation:(12)$$\dot{\underline{x}}=\left(A\left(\underline{x}\right)-B\left(\underline{x}\right)K\left(\underline{x}\right)\right)\underline{x}.$$

### 2.3. Anticancer Treatment Evaluation Metrics

To assess the effectiveness of drug administration scenarios derived from each optimal control approach applied, specific metrics were used. These metrics offer insights into the capacity of each scenario to control tumor growth. Key measurements include the total doses/quantity of the administered drug ($u_{total}$ in mg/kg—also referred to as the *total control effort*) up to the end of the simulation at $t=t_{f}$, and the maximum weight of the tumor ($w_{max}$ in g) reached during the same timeframe:(13)$$\begin{array}{ccc} u_{total}=\sum_{t=0}^{t_{f}}u\left(t\right) & and & w_{max}=\underset{0\leq t\leq t_{f}}{\max}w\left(t\right) \end{array}.$$
Here, $u\left(t\right)$ denotes the dose of the drug administered as determined by the optimal controller at time *t*, and $w\left(t\right)$ represents the weight of the tumor at time t.

In clinical settings, monitoring mouse weight loss is critical as it indicates treatment side effects and overall health, helping assess treatment tolerability during chemotherapy. However, this study does not take into account weight loss, as the dose has already been optimized to be non-toxic and focuses instead on theoretical simulations as a first step rather than actual clinical observations.

## 3. Results

### 3.1. The Optimal Control Approach Applied in Tumor Growth Inhibition Optimal Chemotherapy Determination

Formulating the inhibition of tumor growth as an optimal control problem not only allows for the study of tumor growth dynamics and the minimization of its size at a given endpoint, but also optimizes the application of the control in such a way that the amount of treatment, e.g., each instant dose of anticancer drug, is minimized. Specifically, by applying the above optimal control approach, optimal doses of the anticancer drug gemcitabine administered intraperitoneally (IP) for five different treatment schedules are considered and explored using the linear and non-linear mathematical models for tumor growth inhibition. Gemcitabine (GEM, 2′,2′-difluorodeoxycytidine, dFdC) is a pyrimidine nucleoside and a cytidine analog that is used in combination chemotherapy for the treatment of non-small cell lung cancer, bladder cancer, and breast cancer, with one of the most important clinical applications being, its use as a first-line treatment as monotherapy in patients with metastatic pancreatic cancer. The linear ARX (3,3) TGI model, as introduced in [31,39,40], and the Simeoni et al.’s TGI model [32], as estimated in [33] using data from gemcitabine-treated mouse-human PDAC xenografts previously generated in our laboratories [34], were utilized along with the LQR and SDRE methods applied here for the optimal chemotherapy determination.

In the first scenario, i.e., scenario 1, gemcitabine was administered daily or continuously at the optimal dosage. The aim is to objectively analyze the immediate and sustained effects of the drug and to investigate how a prolonged drug presence affects tumor growth. The rest of the scenarios, i.e., scenarios 2–5, include drug administration every two, three, five, and seven days, respectively, until the tumor’s eradication is achieved. The purpose of studying such intervals is to determine the trade-offs between dosing frequency, drug efficacy in treating tumors, and ultimately the patient’s comfort. Keeping in mind that the end goal is the application of optimally scheduled chemotherapy using the proposed models in the clinics, it is well known that prolonged time intervals may increase patient comfort and reduce side effects, although they may also impact the drug’s effectiveness, allowing tumor regrowth. For this reason, intervals longer than a week (7 days) were not studied.

The main objective of the optimal control problem is to determine the optimal amount of anticancer drug to be administered until the tumor is eradicated. However, it is particularly important to do this with the minimum “cost”. High doses may lead to acute toxicity in healthy cells and subsequently to severe side effects. To avoid such phenomena, it is necessary to impose “hard” constraints on the system’s control variable  u(t), i.e., the chemotherapy dosage. Therefore, when the optimal dose calculated by the controller exceeds a predefined threshold, the control signal, i.e., the dose actually administered, obeys the following inequality:(14)$$u_{min}\leq u{\left(t\right)\leq u}_{max}.$$
where $u_{min}=0$ and $u_{max}$ represent the minimum and the maximum allowable dose levels, respectively. Given that the toxicity of drugs cannot be implicitly modeled mathematically in the TGI mathematical models used in the present study in order to balance toxicity and the chemo treatment efficacy, $u_{max}$ was set at 5.4 mg for a mouse body weight of approximately 27 g, which translates to a $u_{max}=200$ mg/kg [4,5,6,7,8,9,10,11,24,25,29,30,31,32,33,34,39,40,41,42,43,44,45]. The solutions derived under these constraints may result in “suboptimal” control (i.e., suboptimal anticancer agent dose), as they may not strictly minimize the original objective (i.e., Equations (2) and/or (8)). Under the imposed constraints, this ensures a balance between chemotherapy optimality and toxicity minimization, as well as an increased expectation of application in oncology clinical practice.

It is also important to note that to simulate realistic clinical scenarios, control inputs in each case studied, both in the ARX LQR and the “*augmented*” Simeoni et al.’s TGI SDRE (see sections below) were not initiated from day 0, but from day 19 of the experiments [34]. Additionally, tumor weights below $10^{-3}$ g were deemed negligible and set to zero in all simulations. This threshold was chosen because tumor masses of such small size are often undetectable and might have limited clinical relevance in terms of therapeutic intervention.

### 3.2. Linear Optimal Control for Efficient Tumor Growth Eradication

Through linear state feedback, LQR can achieve closed-loop optimal control of the anticancer drug dose levels while at the same time eliminating the tumor. This was performed using a linear ARX (3,3) model, the parameter values of which were estimated in [39] using the data from gemcitabine-treated human PDAC mouse xenografts previously generated in our laboratories [34]. To employ LQR, the model needs to be converted into a state-space form, and this is feasible by calculating the TGI discrete-time dynamics transfer function $\;H\left(z\right)$. Generally, the transfer function of a system can be transformed into a non-unique state-space representation using a discrete-time realization algorithm (DRA) [36]. The transfer function of the single-agent ARX TGI model of [39] was calculated as follows (see [39,40] for details):(15)$$\begin{array}{c} w\left[k\right]=a_{1}w\left[k-1\right]+a_{2}w\left[k-2\right]+a_{3}w\left[k-3\right]+b_{1}u\left[k-1\right]+b_{2}u\left[k-2\right]+b_{3}u\left[k-3\right],thus\\ H\left(z\right)=\frac{W\left(z\right)}{U\left(z\right)}=\frac{b_{1}z^{-1}+b_{2}z^{-2}+b_{3}z^{-3}}{1-a_{1}z^{-1}-a_{2}z^{-2}-a_{3}z^{-3}}. \end{array}$$
where w[k] is the output of the ARX model, i.e., the tumor’s weight prediction at the discrete-time k, $w\left[k-1\right]$, i≤p≤k are the past tumor’s weight observations and $u\left[k-j\right]$, j≤q≤k are the exogenous inputs, i.e., the anticancer agent dose at each discrete time k, with p≤q. The order of the ARX $\left(p,q\right)\;$ model is defined by the set of parameters $\;\left(p,q\right),$ whose values in this study are $\left(p=3,q=3\right)$. These two parameters depict the number of lags considered for the output and input historical data, respectively. Parameters $a_{i}\in R$ and $b_{j}\in R$ are weights associated with each previous observation and exogenous input, respectively. In the context of this study, w represents the tumor’s weight observations, reflecting the size and growth of the tumor over time. On the other hand, u corresponds to the dosages of chemotherapy administered over time, providing a quantitative measure of the treatment intensity.

The estimated coefficients of the linear discrete-time ARX (3,3) TGI model (15) used in this study are given below [39,40]:$$a_{i}=\left[\begin{array}{c} a_{1}\\ a_{2}\\ a_{3} \end{array}\right]=\left[\begin{array}{c} -2.9653\\ \;\;\;\;2.9456\\ -0.9851 \end{array}\right]\;\mathrm{and}\;b_{j}=\left[\begin{array}{c} b_{1}\\ b_{2}\\ b_{3} \end{array}\right]=\left[\begin{array}{c} \;\;\;6.7573\\ -5.8769\\ -8.7736 \end{array}\right].$$

Among several equivalent state-space forms of the above transfer function, the observable canonical form ensures the observability and controllability of the derived state-space system equivalent representation [36]. Thus, the equivalent state-space representation of (15) is described by the observable controllable state-space discrete-time equations below:(16)$$\underline{x}\left[k+1\right]=A\;\underline{x}\left[k\right]+B\underline{u}\left[k\right]\;y\left[k\right]=C\;\underline{x}\left[k\right]+D\;\underline{u}\left[k\right]$$
where$$\begin{array}{cccc} A=\left[\begin{array}{ccc} -a_{1} & 1 & 0\\ -a_{2} & 0 & 1\\ -a_{3} & 0 & 0 \end{array}\right], & \;\;\;\;B=\left[\begin{array}{c} b_{1}\\ b_{2}\\ b_{3} \end{array}\right], & \;\;\;\;C=\left[\begin{array}{ccc} 1 & 0 & 0 \end{array}\right], & \;\;\;\;D=0 \end{array}$$
with$$\underline{x}\left[k\right]={\left[\begin{array}{ccc} x_{1}\left[k\right] & x_{2}\left[k\right] & x_{3}\left[k\right] \end{array}\right]}^{\;T}.\;\;\;\;\;\;\;\;\;\;\;\;\;\;\;\;\;\;\;\;\;\;\;\;\;\;\;\;$$

The state variables $\underline{x}\left[k\right]$ are a set of internal variables in the linear state-space representation (16) without explicit physical meaning [36]; however, they are determined to capture mathematically the tumor’s (output) dynamic response to chemotherapy (input) [40]. Although the above derived state-space discrete-time system is not stable, it is both observable and controllable [36,40]. As a result, the LQR method has the potential to not only stabilize the above system, but also to optimize its performance under specified criteria. The initial conditions of the above state-space system can be calculated from the output $y\left[k\right]$ (i.e., the tumor’s weight at each discrete time k) and the corresponding input $u\left[k\right]$ (i.e., the anticancer agent dose at each discrete time k). Using the tumor growth observations derived from [34,40] for ${\;w}_{k=0}=y\left[k=0\right]$, $w_{k=1}=y\left[k=1\right]$ and $w_{k=2}=y\left[k=2\right]$, the initial values of the states $\underline{x}\left[k\right]$ of the LTI state-space system (16) are calculated as [36,40] (see particularly the pages 96,97,98 of [40]):(17)$$\underline{x}\left[k\right]=\left[\begin{array}{c} x_{1}\left[0\right]\\ x_{2}\left[0\right]\\ x_{3}\left[0\right] \end{array}\right]=O^{-1}\left[\left[\begin{array}{c} y\left[0\right]\\ y\left[1\right]\\ y\left[2\right] \end{array}\right]-T\left[\begin{array}{c} u\left[0\right]\\ u\left[1\right]\\ u\left[2\right] \end{array}\right]\right]=\left[-\begin{array}{c} 0.0200\\ 0.0259\\ 0.0066 \end{array}\right].$$

Based on [36,46], an initial selection was made for the weighting matrices Q and R as $diag\left\{14\;14\;14\right\}\cdot 10^{-4}$ and $343\cdot 10^{2}$, respectively, serving as a “good” start for daily dose administrations [40]. However, for chemotherapy schedules that extend to doses administered every 2, 3, 5, and 7 days, these matrices required adjustments. Consequently, a series of refinements to Q and R were made through a trial-and-error approach to accommodate these extended dosing intervals:$$\begin{array}{ccc} Q=diag\left\{42\;42\;42\right\}\cdot 10^{-4} & and & R=1700. \end{array}$$

To employ LQR control, an optimal feedback gain must be computed. For this purpose, the MATLAB’s (version R2019b) “*dlqr*” function [47] was used to design (calculate) the optimal state-feedback controller, i.e., the optimal chemotherapy schedule. It calculates the optimal LQR gain matrix by minimizing the above quadratic cost function $J_{LQR}\;$(2), representing the trade-off between minimized control effort (i.e., minimized dosage) and the LTI state-space system performance to eradicate the tumor. The optimal feedback gain matrix K as derived by MATLAB, is shown below:$$K=\left[-0.1394,-0.1165,-0.0956\right].$$

#### ARX and LQR-Based Gemcitabine Dosage Optimization for Periodic Tumor Treatment Scenarios

Several cases of different periodic treatment schedules were examined. Specifically, optimal doses of gemcitabine IP for five different treatment schedules were explored. In the first case, i.e., Case A.1, the controller calculated doses for continuous (i.e., everyday) drug administration. For the rest of the cases, i.e., Cases A.2, A.3, A.4, and A.5, the controller calculated optimal drug doses for periodic treatments (see Table 1). Gemcitabine was administered as calculated by the LQR every two, three, five, and seven days for Cases A.2, A.3, A.4, and A.5, respectively, until the tumor’s eradication was achieved. The response of the state-space system (i.e., the tumor’s growth in g) along with the optimal control input (i.e., the gemcitabine dose levels in mg/kg calculated by the LQR) for each case is shown in Figure 1, Figure 2, Figure 3, Figure 4 and Figure 5. The simulation duration varied across different scenarios: for Cases A.1 to A.3, the final time ($t_{f}$) was set at 100 days, while in Case A.4, $t_{f}$ extended to 150 days, and in Case A.5, the simulation ran for a longer period of 300 days.

To better understand the proposed optimal treatment schedules and to evaluate their effectiveness in controlling tumor growth, descriptive statistics were also calculated and are presented in Table 1. $u_{total}$ denotes the cumulative drug intake (i.e., the total dose administered), while $w_{max}$ represents the maximum tumor weight observed during the simulation period. In addition, $w_{zero}$ represents the time duration, in days, required for tumor eradication, indicating the efficacy of each treatment in eliminating the tumor.

In all cases, the chemotherapy treatment starts on day 19, with dose administrations ranging from 130 to 140 mg/kg. Then, the estimated dose levels gradually decreased until the tumor was completely eradicated as late as day 300 and beyond in Case A.5. In Case A.4, shown in Figure 4, noticeable fluctuations in the suggested doses by the LQR and consequently in the tumor’s weight are observed, starting high, below the predefined upper limit of 200 mg/kg, and then fluctuating until the tumor was eradicated. In contrast, Case A.5, shown in Figure 5, starts with a high dosage that steadily decreases until the tumor is completely eradicated. In addition, it is observed that as the intervals between the doses administration increase, both the cumulative amount of drug $u_{total}\;$ required to suppress the tumor’s growth and the duration (days) of the treatment until the tumor’s eradication to$\;w_{zero}$, increase. These increases in the treatment duration particularly appeared when the intervals between the doses administered exceeded 3 days, as observed in cases A.4 and A.5. It is also worth mentioning that in none of the cases did the weight of the tumor exceed 0.75 g.

### 3.3. Non-Linear Optimal Control for Efficient Tumor Growth Eradication

Despite this, the ARX TGI model and its state-space representation, along with LQR, manage to eradicate the tumor. It is important to compare it with more complex dynamic models that also incorporate the PKs–PDs (pharmacokinetics–pharmacodynamics) of the administered drugs. To determine the optimal dose level of gemcitabine, the Simeoni et al. TGI model [32], as estimated in [33] using the data extracted from [34], was further extended. Specifically, an additional state q(t), corresponding to a one-compartment PK-PD mathematical model, was included for the first time in the original three-compartment transit TGI model initially proposed by Simeoni et al. (see [32,33]), thus obtaining the “*augmented*” Simeoni et al.’s TGI non-linear state-space equations model below:(18)$$\begin{array}{c} \frac{dz_{0}\left(t\right)}{dt}=\frac{\lambda_{0}\cdot z_{0}\left(t\right)}{{\left[1+{\left(\frac{\lambda_{0}}{\lambda_{1}}\cdot w\left(t\right)\right)}^{\psi }\right]}^{\frac{1}{\psi }}}-k_{2}\cdot \frac{q\left(t\right)}{V}\cdot z_{0}\left(t\right)\\ \frac{dz_{1}\left(t\right)}{dt}=k_{2}\cdot \frac{q\left(t\right)}{V}\cdot z_{0}\left(t\right)-k_{1}\cdot z_{1}\left(t\right)\\ \frac{dz_{2}\left(t\right)}{dt}=k_{1}\cdot \left[Z_{1}\left(t\right)-Z_{2}\left(t\right)\right]\\ \frac{dz_{3}\left(t\right)}{dt}=k_{1}\cdot \left[z_{2}\left(t\right)-z_{3}\left(t\right)\right]\\ \frac{dq\left(t\right)}{dt}=-k_{10}q\left(t\right)+u\left(t\right)\\ w\left(t\right)=z_{0}\left(t\right)+z_{1}\left(t\right)+z_{2}\left(t\right)+z_{3}\left(t\right). \end{array}$$
In the above non-linear model, the state variables $\left[z_{0}\left(t\right),\;z_{1}\left(t\right),\;z_{2}\left(t\right),\;z_{3}\left(t\right)\right]$ represent key components of the tumor’s dynamic response to chemotherapy over time. Specifically:

$z_{0}\left(t\right)$: represents the proliferating portion of tumor cells, which actively contribute to the tumor’s growth. This state reflects the cells that remain unaffected by the drug’s cytotoxic effects at time t.$z_{1}\left(t\right),\;z_{2}\left(t\right),\;z_{3}\left(t\right)$: represent the tumor’s cells in progressive stages of damage due to the chemotherapy drug. These compartments capture the delayed effect of the drug, where damaged cells are passing through increasingly severe states of damage, before eventually dying.

In the above equations, $k_{1}$ (0.23810 1/day) is the first-order rate constant of the tumor’s growth transit, $k_{2}$ ($4.22\;e^{-4}$ mL/ng ∙ day) is the measure of anticancer drug potency, $\lambda_{0}$ (0.14202 1/day) and $\lambda_{1}$ (0.07618 g/day) are the first- and zero-order rate constants of tumor’s growth (i.e., they characterize the rate of exponential and linear growth), respectively. The $u\left(t\right)$ is the chemotherapy drug dose in mg/kg (and not the drug’s concentration, as is the case in the initially proposed by Simeoni et al.’s TGI model [32]); thus, it is *the control input* at each time instance t, while $w\left(t\right)$ is the corresponding tumor’s weight, i.e., the non-linear model’s (18) *output*. Given that the input $u\left(t\right)$ of Equations (18) is the anticancer drug dosage, $u\left(t\right)\;$ is scalar and ∈R. The initial conditions (at t=0) of the augmented mathematical model (18) are $\left[z_{0}\left(0\right),\;z_{1}\left(0\right),\;z_{2}\left(0\right),\;z_{3}\left(0\right),\;q\left(0\right)\right]=\left[0,0,0,0,0\right]$.

The aim of chemotherapy is to use the appropriate drug doses during treatment to bring the system to a tumor-free equilibrium point $\left[z_{0},\;z_{1},\;z_{2},\;z_{3},\;q\right]=\left[0,0,0,0,0\right]$, where the tumor mass is reduced to zero. In the case of the initially introduced Simeoni et al.’s TGI model this equilibrium is located at the origin; thus, there is no need to employ error states to shift the equilibrium point.

Due to the non-linear nature of the problem, the State-Dependent Coefficient (SDC) form was used to capture the system non-linearities into a *pseudo-linear system matrix*, which is then used to derive the SDRE-based optimal control dosage for tumor growth inhibition, according to [37,38]. The non-linear state-space TGI Equations (18) must be factorized into a SDC form of the *pseudo-linear* form:(19)$$\dot{\underline{x}}=A_{s}\left(\underline{x}\right)\underline{x}+B_{S}\left(\underline{x}\right)\underline{u}$$
where $\underline{x}={\left[x_{1},x_{2},x_{3},x_{4},x_{5}\right]}^{T}={\left[z_{0},z_{1},z_{2},z_{3},q\right]}^{T}$ is the state vector, $\underline{u}(t)\geq 0$ is the input (i.e., the anticancer agent gemcitabine IP dose levels), while $A_{s}\left(\underline{x}\right){\in R}^{5\times 5}$ and $B_{s}\left(\underline{x}\right){\in R}^{5\times 1}$ matrices are the pseudo-linear system matrices in the SDC form.

The initial values of the states at t=0 for (19) are $\left[x_{1}\left(0\right),\;x_{2}\left(0\right),\;x_{3}\left(0\right),\;x_{4}\left(0\right),\;x_{5}\left(0\right)\right]=\left[0,0,0,0,0\right]$. However, the SDC parameterization is not unique, and to preserve the dependency of terms that contain two or more states, *free design* parameters (θ) are introduced. Specifically, a set of vectors of different values of $\theta_{i}\in \left[0,1\right]$, for i=1,2 was used to identify the vector $\underline{\theta }=\left[\theta_{1},\theta_{2}\right]$ that maximizes the pointwise controllable space [38,40]. The SDC parameterization of the non-linear state-space equations system (18) by using the free design parameters $\underline{\theta }$ is described through the algebraic Equations (20), below:(20)$$A_{s}\left(\underline{x},\underline{\theta }\right)=\left[\begin{array}{ccccc} a_{11} & 0 & 0 & 0 & a_{15}\\ a_{21} & a_{22} & 0 & 0 & a_{25}\\ 0 & a_{32} & a_{33} & 0 & 0\\ 0 & 0 & a_{43} & a_{44} & 0\\ 0 & 0 & 0 & 0 & a_{55} \end{array}\right]and{\;B}_{S}=\left[\begin{array}{c} 0\\ 0\\ 0\\ 0\\ 1 \end{array}\right]$$
where
$$\begin{array}{c} a_{11}=\frac{\lambda_{0}}{{\left[1+{\left(\frac{\lambda_{0}}{\lambda_{1}}\cdot w\left(t\right)\right)}^{\psi }\right]}^{\frac{1}{\psi }}}-\left(1-\theta_{1}\right)k_{2}\frac{x_{5}}{V},\\ a_{15}=-\theta_{1}k_{2}x_{1}\frac{1}{V},\\ a_{21}=\left(1-\theta_{2}\right)k_{2}\frac{x_{5}}{V},\\ a_{22}=-k_{1},\\ a_{25}=\theta_{2}k_{2}x_{2}\frac{1}{V},\\ \begin{array}{cc} a_{32}=k_{1}, & a_{33}=-k_{1},\\ a_{43}=k_{1}, & a_{44}=-k_{1}, \end{array}\\ a_{55}=-k_{10}. \end{array}$$

For the given parameter set of Simeoni et al.’s TGI model, presented in [32,33,40], the largest value of $\left|\det\left(M_{c}\right)\right|$ was obtained by choosing through trial-and-error experiments the vector $\underline{\theta }=\left[\theta_{1},\theta_{2}\right]=\left[1,1\right]$, which ensures that the pair ${\{A}_{S}\left(\underline{x},\underline{\theta }\right)$, $B_{s}\}$ is pointwise controllable and, therefore, stabilizable. The SDRE controller for the *augmented* Simeoni et al.’s TGI model, estimated as described above, was designed and developed/programmed in MathWorks MATLAB and SIMULINK (version R2019b) [47]. The goal of non-linear optimal regulation using the SDRE method is to drive all system states to the tumor-free equilibrium at the lowest cost. In other words, it aims to eradicate the tumor by administering minimal doses of the anticancer agent gemcitabine. Thus, the SDRE controller was designed to minimize a cost functional $J_{SDRE-TGI}:$(21)$$J_{SDRE-TGI}=\frac{1}{2}\int_{0}^{\infty }\left({\underline{x}}^{T}Q_{s}\left(\underline{x}\right)\underline{x}+R_{s}\left(\underline{x}\right)u^{2}\right)dt,$$
where $Q_{s}$ and $R_{S}$ are the state and input weighting matrices. Their values were selected through trial-and-error experiments using the experimentally estimated as “good” initial values as:$$\begin{array}{ccc} Q_{S}=diag\left(1\cdot e^{3},\;1\cdot e^{3},\;1{\cdot e}^{3},\;1{\cdot e}^{3},\;0.04\right) & and & R_{s}=25 \end{array}.$$

#### 3.3.1. Optimized Chemotherapy Dosages in Periodic Tumor Treatment Scenarios Using SDRE Optimal Control and the Introduced Here “Augmented” Simeoni et al.’s TGI Non-Linear State-Space Dynamic Model

Optimum gemcitabine dosages and the corresponding metrics, such as $u_{total}$ and $w_{max}$, for each case explored, are listed in Table 2. This table also serves as a reference point for comparing the results of the different treatment regimens, thereby aiding in the decision-making process when selecting the most efficient and effective treatment plan. The suboptimal drug delivery protocols related to the SDRE control for the periodic cases are depicted in Figure 6, Figure 7, Figure 8, Figure 9 and Figure 10 [40].

In all treatment schedules, the optimal doses of gemcitabine proposed by the controller are highly effective in reducing tumor growth. Specifically, in Cases B.1–B.3, the tumor weight reached values on the scale of $1\cdot e^{-2}$ g or less. Such values are below the tumor’s weight $w_{0}$ at the time of inoculation t=0. In Cases B4 and B5, the tumor’s weight reached values almost equal to $w_{0}.$ Thus, they can be described almost as undetectable. However, it was observed that the drug dosages suggested by the SDRE were not zero after the 60th or 80th days, respectively. This observation suggests that the controller may have tried to maintain the tumor at a low level rather than eradicating it completely. This is the case in each periodic treatment schedule explored. The tumor weight is rapidly reduced, and it is stabilized to a certain weight of $1\cdot e^{-2}$ g at day 60 of the simulations for Cases B1 and B2 (i.e., day 60 of the anticancer drug administration) and of $1\cdot e^{-1}$ at day 80 for Cases B3, B4, and B5 (i.e., day 80 of the anticancer drug administration). Thus, the anticancer agent administration could be stopped after the 60th day of administration for Cases B1 and B2 and after the 80th day of administration for Cases B3 and B4 to reduce significantly the total administrated agent, i.e., to reduce $u_{total}$ about 2200 mg/kg–2400 mg/kg in all the above cases. Such behavior could result from the objective function used to optimize drug doses or even from the selection of the state-dependent weighting matrices Q and  R heuristically. Another potential factor may be the dose administration intervals. Longer intervals could act as rest periods, allowing the tumor to recover slightly and thus requiring subsequent doses. The maximum tumor reached is lower than 0.5 g. Nonetheless, as the interval of the doses increases, an increase in the maximum tumor weight is observed during the simulations. This increase in the maximum tumor’s weight, $w_{max}$, observed when the dosing intervals become longer may confirm this hypothesis. In all cases, the suggested doses start aggressively at the onset of the treatment, beginning with doses of 200 mg/kg, reaching the predefined upper dose limit. As the drug takes effect on the tumor, the SDRE controller suggests progressively lower doses. To determine whether non-zero drug doses are necessary to maintain the desired tumor size, future work could involve modifying the objective function and further optimizing the SDRE controller to reduce drug doses while maintaining the tumor eradication control. Overall, the results demonstrate the effectiveness of the optimal drug doses proposed by the SDRE control method in reducing tumor growth to low levels.

#### 3.3.2. Optimized Chemotherapy Dosages in Intermittent Tumor Treatment Cases Using the Introduced Here “Augmented” Simeoni et al.’s TGI Non-Linear State-Space Dynamic Model and SDRE Optimal Control

Whereas in the above-mentioned cases, the tumor was successfully eliminated, chemotherapy-resistant scenarios might occur due to the long exposure to the drug. Cancer chemotherapy resistance is a phenomenon where the neoplastic cells develop the ability to evade the effects of the chemotherapeutic treatment, leading to failure in drug response [41,48]. In this regard, optimal drug dosage determination for intermittent chemotherapy schedules is also valuable to be explored; thus, it is proposed and its efficiency is investigated in the present work. To be more precise, optimal dosages for two cases of different intermittent treatment schedules were investigated. In both cases, chemotherapy was administered every three days for a total of five times, i.e., q3dx5. To mitigate toxicity and drug resistance, the treatment is then paused for a period of ${tr}_{p}=7$ (Case B.6) and 10 days (Case B.7), respectively. Following the treatment pause (i.e., chemotherapy’s “holiday”), the treatment is resumed with dose administrations every five days for a total of five cycles, denoted as q5dx5. The above process is repeated until the end of the simulation, i.e., until the end of the treatment period ($t_{f}=100$ days).

A summary of the results is presented in Table 3 below, while the tumor weight growth curves and the optimal drug dosages are illustrated in Figure 11 and Figure 12. In examining these results, a striking observation is the efficacy of the intermittent SDRE treatment schedules in both cases B.6 and B.7. Over approximately 100 days, the tumor size was stabilized at around 0.067 g, a weight close to the initial $w_{0\;}$ at the inoculation time for Case B.6, while for Case B.7 it was at around 0.089 g, demonstrating the potential of these treatment strategies. The proposed drug doses in both cases start aggressively, with initial doses close to 200 mg/kg, an average of 156 mg/kg in Case B.6, and an average of 170 mg/kg in Case B.7. A comparison between the two cases reveals a direct correlation between the length of the chemotherapy “holidays” denoted by ${tr}_{p}$, and the total amount of drug administered, $u_{total}$. Specifically, Case B.7, which had longer chemotherapy “holidays”, required a 3.34% lower total amount of drug than Case B.6. Moreover, this total amount of drug reduction has also resulted in faster tumor eradication, potentially reducing the chemotherapy’s toxicity.

## 4. Discussion

Anticancer chemotherapy is an intensive systemic treatment where drugs circulate through the bloodstream, affecting both cancerous cells at the tumor site and healthy cells. Consequently, prolonged exposure, particularly to high drug concentrations, can result in acute toxicity and side effects. Utilizing accurate mathematical modeling of the tumor’s growth and optimal control theory could provide a very promising solution to this challenge. Thus, two well-established optimal control methods, LQR and SDRE, have been successfully used in combination with TGI mathematical models to determine optimal dose regimens that can reduce tumor size while minimizing side effects. In the first approach, the non-linear nature of the Simeoni et al.’s PK-PD TGI model [32] makes the SDRE optimal non-linear control method particularly suitable. To apply the SDRE methodology, the non-linear TGI model had to be converted to a pseudo-linear form. However, before the model’s transformation to the pseudo-linear form, its state vector was expanded (augmented) for the first time to include a one-compartment PK model, with the goal of capturing more accurately the PKs of the administered drug doses (gemcitabine IP in the scenario examined). In the second approach, the LQR method was tested along with an ARX model, as estimated in [31,39], to determine optimal gemcitabine regimens. In this later approach, it was essential to transform the ARX model into an equivalent state-space representation.

In both cases, periodic and intermittent optimized treatment schedules were extensively explored and compared for the first time. To mitigate excessive toxicity, strict limits were set on the doses of drugs suggested by the optimal controllers. In all the scenarios studied, including continuous, periodic, and intermittent administration of the drug with treatment pauses, the size of the tumor was reduced. It is shown that the introduction of intermittent treatment schedules could serve as a potential alternative, with the aim of minimizing toxicity while eradicating efficiently the tumor and thus improving the patient’s overall quality of life during the treatment period.

In the cases of ARX TGI and LQR, the tumor weight was grounded to zero., i.e., the tumor was eliminated. On the other hand, when utilizing the non-linear TGI model along with the SDRE controller, the weight of the tumor was reduced, but not eliminated. Instead, it stabilized at a small value around the tumor weight at the time of inoculation. Moreover, the SDRE controller suggested optimal dose levels higher than those calculated by the LQR for similar treatment scenario case studies. The difference in methods, as well as the simplification in modeling tumor growth dynamics with a linear model such as ARX vs. the non-linear one of Simeoni et al., may be the cause of these differences in results (i.e., tumor weight eradication and proposed optimal doses). Nevertheless, prolonged treatments with high dosage levels may result in severe side effects for the patient.

The methods presented in this work can provide oncologists with computational tools to design, at no cost, optimal and patient-specific chemotherapy schedules to successfully fight cancer while improving the quality of life of patients during treatment. Studies have shown that mCHT (metronomic chemotherapy, i.e., continuous or frequent administration of low-dose chemotherapy drugs) may be a promising strategy to control tumor growth more effectively [42,43,49]. In addition, the emergence of artificial intelligence (AI) and machine learning (ML) techniques [43] offers new ways to further optimize mCHT schedules, personalized to each patient’s unique profile, also in combination with methods such as those presented here, to increase the success of therapy, and should be a focus of future investigations. Moreover, all the relevant methodologies can also be used to achieve higher efficiency while reducing the cost of the new anticancer agent’s research and development. Ultimately, as our understanding of the intricate dynamics of cancer and the potential of multi-drug regimens advances, comprehensive research efforts to identify optimal treatment strategies will remain critical in the ongoing fight against this very complex, persistent, and highly lethal for humanity disease.

Although we do not use clinical data in the current paper, but rather data from preclinical studies using human-to-mouse PDAC xenografts, we must highlight several novelties that have emerged: (i) For the first time, the simplified linear autoregressive model with exogenous inputs (ARX) directly relates the dynamics of tumor growth inhibition to the dose of the anticancer agent, in which the PK-PD of the drug is not considered explicitly but implicitly in the identified parameters of the linear model, obtained by using chemotherapy dose (input) vs. tumor weight (output) data derived from human-to-mouse xenografts; (ii) For the first time, “bang-bang” linear quadratic control theory, based on an ARX linear mathematical model identified using preclinical laboratory data (real measurements), is used to determine the best chemotherapy schedule to most effectively inhibit tumor growth; (iii) For the first time, the “extended” non-linear mathematical model of Simeoni et al. TGI is presented. This non-linear TGI model incorporates the PK-PD drug dynamics into the proposed state-space representation, allowing the determination of the optimal anticancer drug dose directly in mg/kg, and not in the units of its “concentration in the blood” [4,5,6,29,32]. This concentration is the most commonly calculated in all similar studies but is very difficult to translate into the corresponding mg/kg dose, which is usually administered in clinical practice; (iv) For the first time, in addition to periodic, intermittent anticancer drug administration, therapeutic scenarios were established, and their efficiency was evaluated by simulation using the “extended” mathematical model TGI of Simeoni et al. and pseudo-linear (SDRE) control theory and results; (v) An easy-to-implement, efficient, useful, and valuable computer-assisted tool in cancer chemotherapy is given and used successfully.

The importance of exploring optimal treatment regimens extends beyond chemotherapy alone. There is a growing interest in combination therapies, such as chemo-immunotherapy, which have shown great efficacy [4,5,6,7,8,9,10,11,32,39]. In addition, continued efforts to integrate advanced mathematical modeling with advanced optimal control and/or AI-ML strategies, including LQR and SDRE, are essential and may lead to improvements in drug dosing strategies for cancer treatment. It is important to note that although this work is primarily based on hypothetical scenarios and computer-simulated case studies, animal (preclinical) studies are currently underway to validate and confirm the true predictability of these models and methods. In conclusion, although more studies are needed in this direction, ultimately, the selection of anti-tumor drug doses for each patient in clinical practice could be significantly and more efficiently improved by the proper integration of similar mathematical/computational approaches, such as those described herein, which are considered to be useful and valuable computational tools in oncology.

## Figures and Tables

**Figure 1 biomedicines-13-00315-f001:**
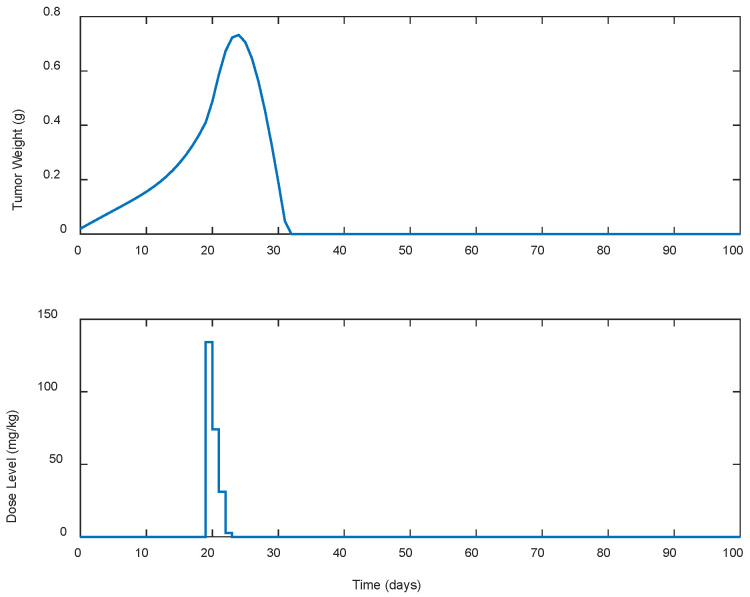
ARX system’s output (i.e., tumor weight) and optimal control input (i.e., dose level). Case A.1: Daily dose administration starting on day 19.

**Figure 2 biomedicines-13-00315-f002:**
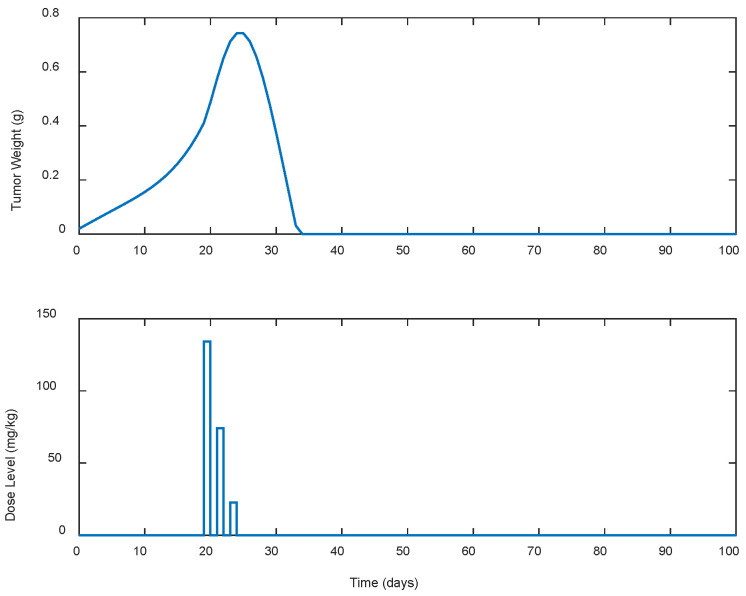
ARX system’s output (i.e., tumor weight) and optimal control input (i.e., dose level). Case A.2: Periodic treatment with dose administration every 2 days starting on day 19.

**Figure 3 biomedicines-13-00315-f003:**
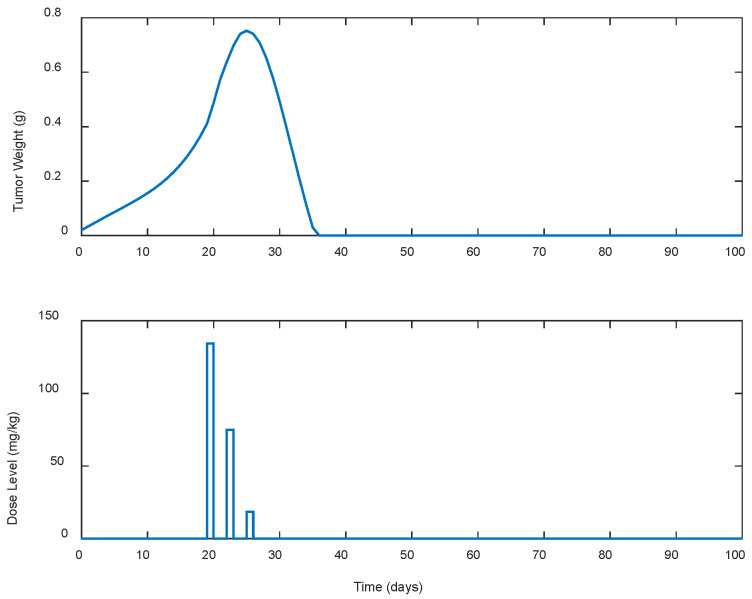
ARX system’s output (i.e., tumor weight) and optimal control input (i.e., dose level). Case A.3: Periodic treatment with dose administration every 3 days starting on day 19.

**Figure 4 biomedicines-13-00315-f004:**
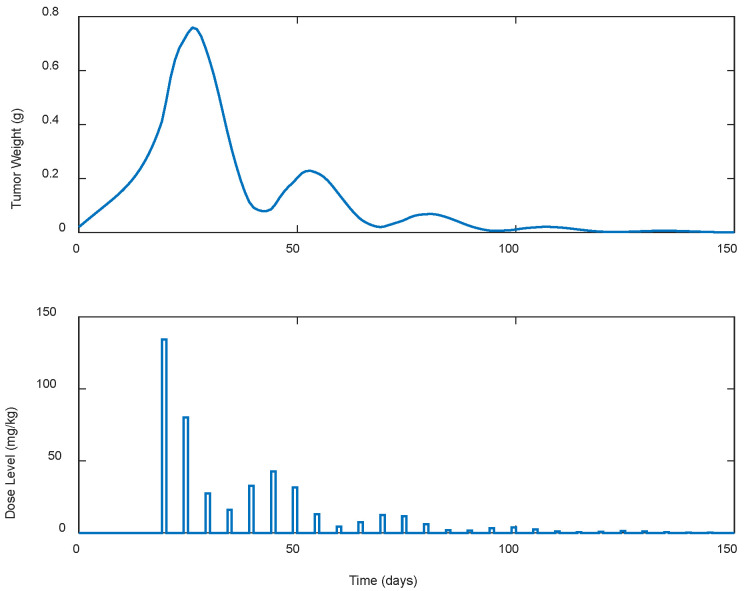
ARX system’s output (i.e., tumor weight) and optimal control input (i.e., dose level). Case A.4: Periodic treatment with dose administration every 5 days starting on day 19.

**Figure 5 biomedicines-13-00315-f005:**
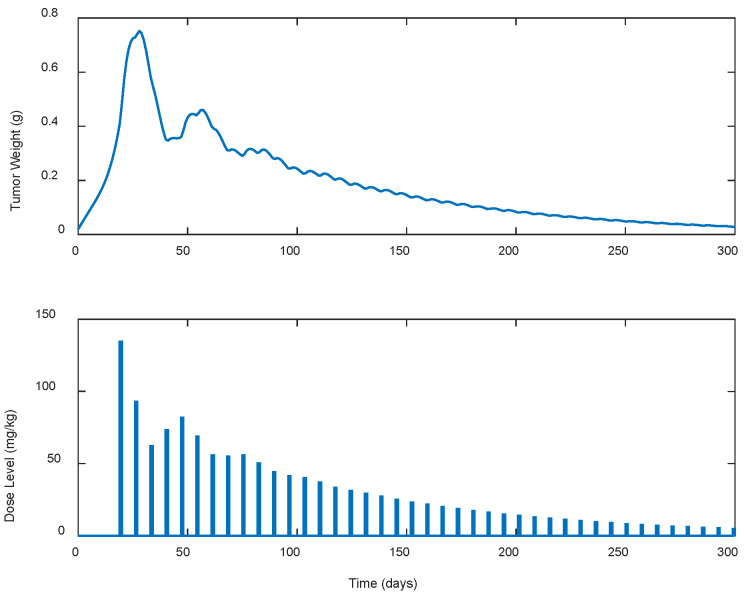
ARX system’s output (i.e., tumor weight) and optimal control input (i.e., dose level). Case A.5: Periodic treatment with dose administration every 7 days starting on day 19.

**Figure 6 biomedicines-13-00315-f006:**
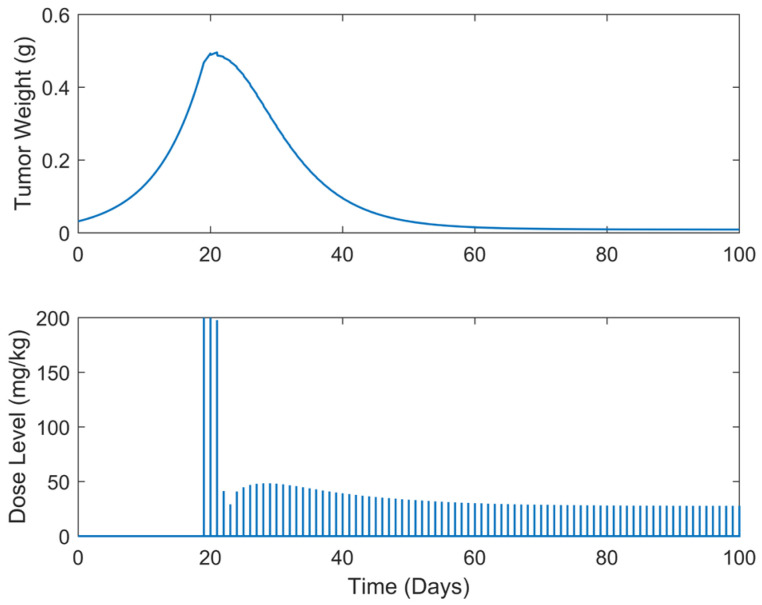
“Augmented” TGI system’s output (i.e., tumor weight) and optimal control input (i.e., dose level). Case B.1: Daily dose administration days starting on day 19.

**Figure 7 biomedicines-13-00315-f007:**
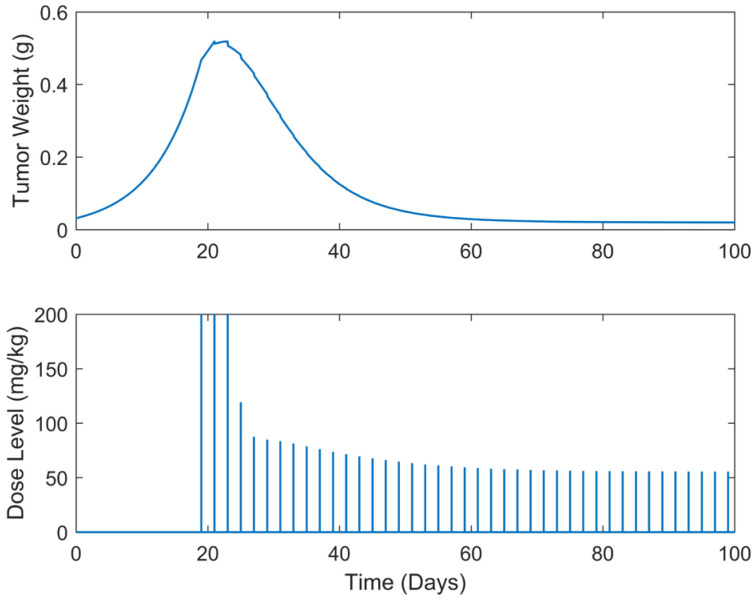
“Augmented” TGI system’s output (i.e., tumor weight) and optimal control input (i.e., dose level). Case B.2: Periodic treatment with dose administration every 2 days starting on day 19.

**Figure 8 biomedicines-13-00315-f008:**
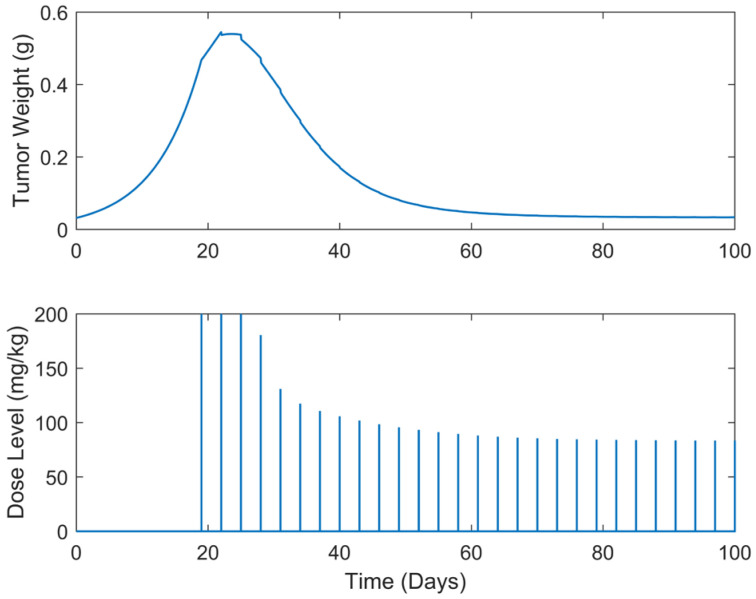
“Augmented” TGI system’s output (i.e., tumor weight) and optimal control input (i.e., dose level). Case B.3: Periodic treatment with dose administration every 3 days starting on day 19.

**Figure 9 biomedicines-13-00315-f009:**
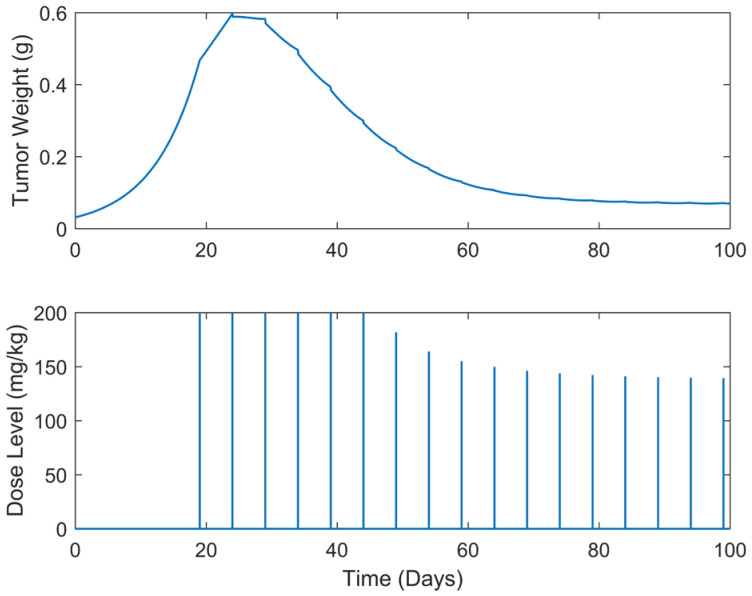
Augmented” TGI system’s output (i.e., tumor weight) and optimal control input (i.e., dose level). Case B.4: Periodic treatment with dose administration every 5 days starting on day 19.

**Figure 10 biomedicines-13-00315-f010:**
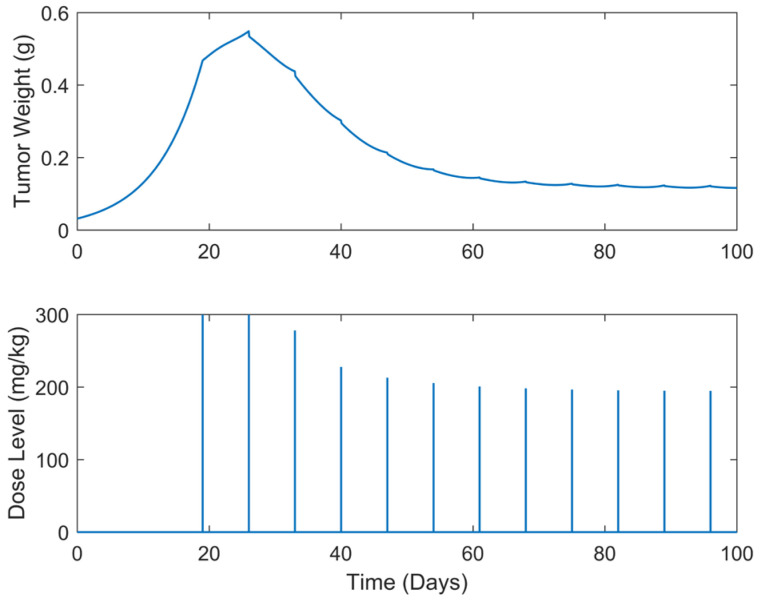
Augmented” TGI system’s output (i.e., tumor weight) and optimal control input (i.e., dose level). Case B.5: Periodic treatment with dose administration every 7 days starting on day 19.

**Figure 11 biomedicines-13-00315-f011:**
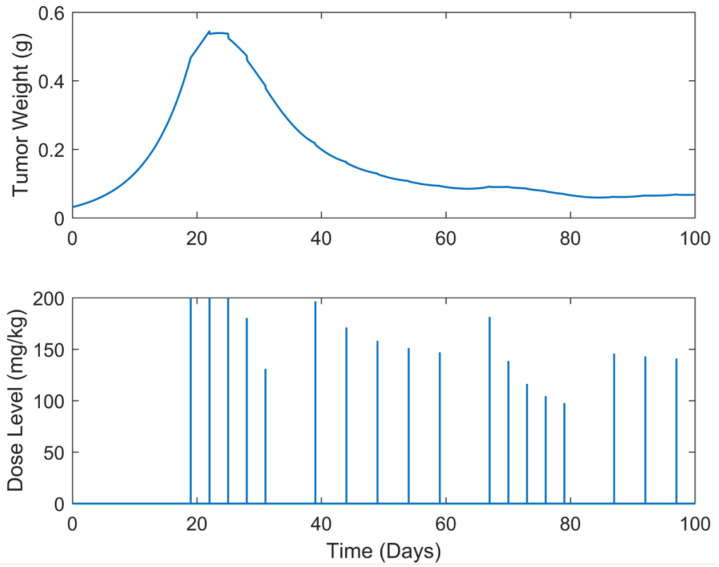
“Augmented” TGI system’s output (i.e., tumor weight) and optimal control input (i.e., dose level). Case B.6: Intermittent treatment of q3dx5 and q5dx5 dose administrations separated with $tr_{p}=7$-day pauses, starting on day 19.

**Figure 12 biomedicines-13-00315-f012:**
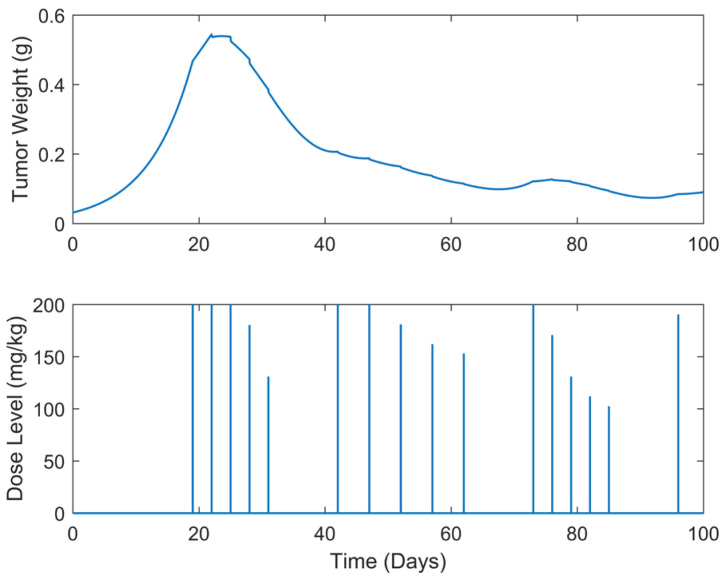
“Augmented” TGI system’s output (i.e., tumor weight) and optimal control input (i.e., dose level). Case B.7: Intermittent treatment of q3dx5 and q5dx5 dose administrations separated with $tr_{p}=10$-day pauses, starting on day 19.

**Table 1 biomedicines-13-00315-t001:** Periodic LQR optimal control treatment results across Cases A.1 to A.5. Anticancer agent dose schedules and metrics.

Case	Dose Schedule	$u_{total}$ (mg/kg)	$w_{max}$ (g)	$w_{zero}$ (Days)
**A.1**	continuous	241.93	0.7325	33
**A.2**	every 2 days	230.96	0.7428	35
**A.3**	every 3 days	227.59	0.7522	37
**A.4**	every 5 days	437.21	0.7591	148
**A.5**	every 7 days	1296.45	0.7523	>300

**Table 2 biomedicines-13-00315-t002:** Periodic SDRE optimal control treatment results across Cases B.1 to B.5. Anticancer agent dose schedules and metrics.

Case	Dose Schedule	$u_{total}$ (mg/kg)	$w_{max}$ (g)	$w_{zero}$ (Days)
**B.1**	continuous	3234.63	0.4957	>100
**B.2**	every 2 days	3062.82	0.5188	>100
**B.3**	every 3 days	3002.18	0.5447	>100
**B.4**	every 5 days	2844.42	0.5970	>100
**B.5**	every 7 days	2705.71	0.5486	>100

**Table 3 biomedicines-13-00315-t003:** Intermittent SDRE optimal control treatment results across Cases B.6 to B.7. Anticancer agent dose schedules and metrics.

Case	Dose Schedule	$tr_{p}$ (Days)	$u_{total}$ (mg/kg)	$w_{max}$ (g)	$w_{zero}$ (Days)
**B.6**	q3dx5, q5dx5	7	2804.91	0.5447	>100
**B.7**	q3dx5, q5dx5	10	2714.38	0.5447	>100

## Data Availability

Data are contained within the article.

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
