# Peer review of "Linear and Non-Linear Optimal Control Methods to Determine the Best Chemotherapy Schedule for Most Effectively Inhibiting Tumor Growth"

_biomedicines, 2025, doi:10.3390/biomedicines13020315_

Round 1

Reviewer 1 Report

Comments and Suggestions for Authors

The problem, while interesting at first glance, must be treated as a purely theoretical work with little translation into practical applications even when authors claims the opposite. The paper completely ignores the problem of pharmacokinetics and pharmacodynamics of drugs in the first model and uses very simple PK-PD in the second. The toxicity of drugs and methods of dealing with it were omitted, assuming only that the dose should be minimized. Meanwhile, the above aspects are real problems limiting chemotherapy.

Detailed comments.

1. Authors mention gemcitabine in line 172 but do not explaining what it is and how it works before.

2. Starting from line 193 to 204 it seems that authors wants to recreate some real experiments done on mice but none is described in the introduction part so it is very confusing.

3. The choice of the models used is not well explained. There is plenty of tumor grow models available, even with the toxicity, PK and PD included. Why not use them instead?

4. In line 363 we have some compilation error saying: "Error! Reference source not found."

Author Response

The problem, while interesting at first glance, must be treated as a purely theoretical work with little translation into practical applications even when authors claims the opposite. The paper completely ignores the problem of pharmacokinetics and pharmacodynamics of drugs in the first model and uses very simple PK-PD in the second. The toxicity of drugs and methods of dealing with it were omitted, assuming only that the dose should be minimized. Meanwhile, the above aspects are real problems limiting chemotherapy.

We thank the reviewer for carefully reading and reviewing our manuscript and providing valuable comments. We fully agree with the reviewer. In our manuscript, we present theoretical work that is partly based on in vivo preclinical data from our previous work (see reference 39). We describe a simple mathematical theoretical model that aims to computationally determine the best chemotherapy schedule to most effectively inhibit tumor growth. This is clearly mentioned in several points in the paper. To this end, we have used a drug (gemcitabine) with known PK and PD data. The methods and approaches presented here are a theoretical "computational tool" in the field of oncology, but of course, the reviewer is right and to build a more robust predictive model, real PK and PD data need to be considered, especially in the context of drug discovery and development. However, the innovative results and achievements of this work increase the expectation of its application to clinical practice in oncology.

Detailed comments.

  1. Authors mention gemcitabine in line 172 but do not explaining what it is and how it works before.

We thank the reviewer for pointing out this omission. It has been addressed (see page 5, lines 187-191)

  1. Starting from line 193 to 204, it seems that authors want to recreate some real experiments done on mice but none is described in the introduction part so it is very confusing.

As reported in the manuscript, the data of chemotherapy dose (input) versus tumour weight (output) are derived from a previous preclinical study performed in our laboratory using human-to-mouse PDAC xenografts.

  1. The choice of the models used is not well explained. There is plenty of tumor grow models available, even with the toxicity, PK and PD included. Why not use them instead?

As mentioned in our response to this reviewer's comment 2, the data used were derived from previous real in vivo preclinical experiments performed and published by our laboratory (see ref 39).

  1. In line 363 we have some compilation error saying: "Error! Reference source not found."

We thank the reviewer for pointing out this error. We apologize for this inconvenience. It has been corrected.

Reviewer 2 Report

Comments and Suggestions for Authors

This study optimizes chemotherapy schedules using mathematical modeling and optimal control to solve a difficult problem. However, some regions need major improvement: 1-The paper compares linear ARX and nonlinear Simeoni et al. models without discussing applicability. Their clinical relevance and assumptions need clarification. 2-The manuscript lists many studies but ignores important context-enhancing advances. Cite these relevant works in the revised manuscript: A novel modal series representation approach to solve a class of nonlinear optimal control problems, International Journal of Innovative, Computing, Information and Control, 7(3), 1413-1425, 2011; An efficient finite difference method for the timedelay optimal control problems with timevarying delay, Asian Journal of Control, 19(2), 554-563, 2017. 3-The study uses mostly simulated data without clinical or experimental validation. Discussion of clinical translation or experimental validation would strengthen the paper. 4-Tables and Figures 1-10 and 1-3 show relevant data without statistical analysis. Test daily and periodic schedules statistically. 5-Readability and language changes are needed. 6-The article discusses AI/ML chemotherapy schedule optimization. Given the study's limitations, authors should pursue this promising path. As a result, I recommend this paper for publication after the above major revision.

Comments on the Quality of English Language

The English could be improved to more clearly express the research.

Author Response

This study optimizes chemotherapy schedules using mathematical modeling and optimal control to solve a difficult problem. However, some regions need major improvement:

1-The paper compares linear ARX and nonlinear Simeoni et al. models without discussing applicability. Their clinical relevance and assumptions need clarification.

This is, as mentioned at several points in the manuscript, a theoretical paper that presents ideas and mathematical models that may help to improve and lead to a more accurate dosing schedule. As mentioned in lines 90-99 of this study, two tumor growth inhibition (TGI) models, (i) a simplified linear auto-regressive with exogenous inputs (ARX) directly relating tumour growth inhibition and tumour weight at each time point to anticancer drug dose, in which the PK-PD of the drug is not "explicitly" but "implicitly" taken into account, i.e. in the identified parameters of the linear model, numerical values obtained by using the data of chemotherapy dose (input) versus tumour weight (output) from preclinical real experiments that were performed previously in our laboratories using human-to-mouse PDAC xenografts [39]; (ii) the well-known non-linear mathematical model of Simeoni et al.[37, 38] for tumour growth inhibition, which explicitly takes into account the PK-PD of the drug and whose parameters are identified similarly to (i), are combined with optimal control methods, namely the LQR and the SDRE respectively [32-35].  

However, to address the reviewer's comment and to highlight the importance of our work, a new section has been added to the discussion (lines 538-557) that points out the novelties of our work and reads as follows:

“Although we don't use clinical data in the current paper, but rather data from preclinical studies using human-to-mouse PDAC xenografts, we must highlight several novelties that have emerged: (i) For the first time, the simplified linear autoregressive with exogenous inputs (ARX) directly relates the dynamics of tumour growth inhibition to the dose of the anticancer agent, in which the PK-PD of the drug is not considered explicitly but implicitly in the identified parameters of the linear model, obtained by using chemotherapy dose (input) vs tumour weight (output) data derived from human-to-mouse xenografts; (ii) For the first time, "bang-bang" linear quadratic control theory, based on an ARX linear mathematical model identified using preclinical laboratory data (real measurements), is used to determine the best chemotherapy schedule to most effectively inhibit tumour growth. (iii) For the first time, the "extended" non-linear mathematical model of Simeoni et al. TGI is presented. This model incorporates the PK-PD drug dynamics into the proposed state space representation, allowing the determination of the optimal anticancer drug dose directly in mg/kg, which is the most calculated in all similar studies and which is very difficult to translate into the corresponding mg/kg dose to be administered in clinical practice. For the first time, in addition to periodic, intermittent anticancer drug administration therapeutic scenarios were established and their efficiency was evaluated by simulation using the "extended" mathematical model TGI of Simeoni et al. and pseudo-linear (SDRE) control theory and results. (v) An easy-to-implement, efficient, useful, and valuable computer-assisted tool in cancer chemotherapy is given and used successfully.

Further down in lines 567-572, we conclude, “although more studies are needed in this direction, ultimately, the selection of anti-tumour drug doses for each patient in clinical practice could be significantly and more efficiently improved by proper integration of similar mathematical/computational approaches such as those described herein, which are considered to be useful and valuable computational tools in oncology”.

 2-The manuscript lists many studies but ignores important context-enhancing advances. Cite these relevant works in the revised manuscript: A novel modal series representation approach to solve a class of nonlinear optimal control problems, International Journal of Innovative, Computing, Information and Control, 7(3), 1413-1425, 2011; An efficient finite difference method for the time‐delay optimal control problems with time‐varying delay, Asian Journal of Control, 19(2), 554-563, 2017.

We appreciate reviewer's thorough revision of our manuscript. However, the manuscript does not ignore the proposed works to be cited. The two important and valuable works suggested by the reviewer to be cited clearly concern purely theoretical and advanced research results in pure control and automation theory. However, the paper does not carry out any research in control and automation theory. In the paper, well-known linear and "pseudolinear" control theory and the corresponding theoretical results, which are already well established, i.e. see the corresponding references, are successfully applied to computationally determine the best chemotherapy schedule to most effectively inhibit tumour growth. Thus, the two papers suggested by the reviewer to be cited in the paper are not relevant to the material and methods of the paper, its purpose, etc. For this serious reason, the authors believe that these specific references should not be included and discussed in the current paper.

3-The study uses mostly simulated data without clinical or experimental validation. Discussion of clinical translation or experimental validation would strengthen the paper.

Please see response in comment 1.

4-Tables and Figures 1-10 and 1-3 show relevant data without statistical analysis. Test daily and periodic schedules statistically.

It is an internal characteristic of the numerical simulations carried out in the present work to provide accurate (i.e. concrete) values regarding the evolution of the tumour under chemotherapy, without any stochasticity in the obtained results.

5-Readability and language changes are needed.

We have revised the manuscript in terms of language and we hope that we have now addressed the concerns of the reviewer in terms of readability and language issues.

6-The article discusses AI/ML chemotherapy schedule optimization. Given the study's limitations, authors should pursue this promising path.

We do not actually "discuss" AI/ML approaches, but only mention them very briefly in the conclusions of the paper as a call for further research. We certainly agree with the reviewer. We also believe that the incorporation of AI/ML is indeed a very promising way to optimise chemotherapy, and we thank the reviewer for this valuable suggestion, which we will certainly pursue, but at this point we respectfully believe it is beyond the scope of this paper.

As a result, I recommend this paper for publication after the above major revision.

Reviewer 3 Report

Comments and Suggestions for Authors

The authors write:

“Utilizing both linear ARX” 

Comment1: What is ARX? No abbreviations in abstract.

“…and non-linear modeling approaches for tumor growth inhibition, this study explores various single agent treatment protocols, including continuous, periodic, and intermittent chemotherapy schedules.” 

Comment2: In which clinic should this happen? These are just mathematical experiments. No clear connection to any actual treatment scheme in clinic is given.

“By integrating advanced mathematical modeling with optimal control theory and meth-ods, namely the Linear Quadratic Regulator (LQR) and the State-Dependent Riccati Equation  (SDRE), this work explores and assesses effective and patient-centric chemotherapy treatment strategies.”

Comment 3: Again, some comparison to treatment schemes from a typical clinic or intensive care unit (continuous infusion for example) should be done. Once this has been achieved, some of the calculations can be compared to actual data from clinic. Both steps are currently completely missing.

Results claim of the authors: “The integration of these methodologies provides insights into how different administration schedules can impact tumor response. The study assesses the effectiveness of various periodic and intermittent chemotherapy treatment strategies, focusing on optimizing drug dose levels and minimizing potential side effects for patients undergoing chemotherapy”

Comment 4: Again, the value of these theoretical exercises is completely unclear. There should be systematic comparison with real data (either from clinic or from animal experiments) to assess the realism or usefulness of the “treatment” schemes calculated.

Comment 5: Similarly, a number of related theoretical modelling articles are cited, but it is also here not clear how far the authors simply follow directly these earlier articles or what the true novelty of their own approach is.

We read “patient-centric treatment strategies in chemotherapy”: 

Comment 6: Well, this would become believable if at least concrete data from some patients are described and compared to.

Author Response

First of all, we would like to thank the reviewers for their careful and thorough reading of our manuscript and for their valuable comments, which helped us to improve our work.

The authors write:

“Utilizing both linear ARX” 

Comment1: What is ARX? No abbreviations in abstract.

We thank the reviewer for pointing out this omission. It has been addressed (See Abstract, page 1, line 18)

 “…and non-linear modeling approaches for tumor growth inhibition, this study explores various single agent treatment protocols, including continuous, periodic, and intermittent chemotherapy schedules.” 

Comment 2: In which clinic should this happen? These are just mathematical experiments. No clear connection to any actual treatment scheme in clinic is given.

The reviewer is correct, and we apologize for the confusion caused by not properly reporting the experimental model used to generate and retrieve the chemotherapy data. We have corrected this, see lines 90-99 where we report that the data are derived from preclinical in vivo experiments in human-to-mouse xenografts. Obviously, there are no clinical but only preclinical data. 

“By integrating advanced mathematical modeling with optimal control theory and methods, namely the Linear Quadratic Regulator (LQR) and the State-Dependent Riccati Equation (SDRE), this work explores and assesses effective and patient-centric chemotherapy treatment strategies.”

Comment 3: Again, some comparison to treatment schemes from a typical clinic or intensive care unit (continuous infusion for example) should be done. Once this has been achieved, some of the calculations can be compared to actual data from clinic. Both steps are currently completely missing.

Please see our response to comment 2. As already mentioned above we use only preclinical data in a fully theoretical approach.

 Results claim of the authors: “The integration of these methodologies provides insights into how different administration schedules can impact tumor response. The study assesses the effectiveness of various periodic and intermittent chemotherapy treatment strategies, focusing on optimizing drug dose levels and minimizing potential side effects for patients undergoing chemotherapy”

Comment 4: Again, the value of these theoretical exercises is completely unclear. There should be systematic comparison with real data (either from clinic or from animal experiments) to assess the realism or usefulness of the “treatment” schemes calculated.

We agree with the reviewer and such work is planned for our laboratories in the near future, involving more drugs and more dosing regimens. However, as we mentioned the current work is a theoretical work that needs to be validated by more studies (see discussion lines 568-573) “Although more studies are needed in this direction, ultimately, the selection of anti-tumour drug doses for each patient in clinical practice could be significantly and more efficiently improved by proper integration of similar mathematical/computational approaches such as those described herein, which are considered to be useful and valuable computational tools in oncology”. Finally, to point out the value of our work we have added a new section in discussion lines 538-557) that points out the novelties of our work and reads as follows:

Although we don't use clinical data in the current paper, but rather data from preclinical studies using human-to-mouse PDAC xenografts, we must highlight several novelties that have emerged: (i) For the first time, the simplified linear autoregressive with exogenous inputs (ARX) directly relates the dynamics of tumour growth inhibition to the dose of the anticancer agent, in which the PK-PD of the drug is not considered explicitly but implicitly in the identified parameters of the linear model, obtained by using chemotherapy dose (input) vs tumour weight (output) data derived from human-to-mouse xenografts ; (ii) For the first time, "bang-bang" linear quadratic control theory, based on an ARX linear mathematical model identified using preclinical laboratory data (real measurements), is used to determine the best chemotherapy schedule to most effectively inhibit tumour growth. (iii) For the first time, the "extended" non-linear mathematical model of Simeoni et al. TGI is presented. This model incorporates the PK-PD drug dynamics into the proposed state space representation, allowing the determination of the optimal anticancer drug dose directly in mg/kg, which is the most calculated in all similar studies and which is very difficult to translate into the corresponding mg/kg dose to be administered in clinical practice. (iv)For the first time, in addition to periodic, intermittent anticancer drug administration therapeutic scenarios were established and their efficiency was evaluated by simulation using the "extended" mathematical model TGI of Simeoni et al. and pseudo-linear (SDRE) control theory and results. (v) An easy-to-implement, efficient, useful and valuable computer-assisted tool in cancer chemotherapy is given and used successfully.”

Comment 5: Similarly, a number of related theoretical modelling articles are cited, but it is also here not clear how far the authors simply follow directly these earlier articles or what the true novelty of their own approach is.

Please see our response to comment 4 regarding the novelties of our work.

 We read “patient-centric treatment strategies in chemotherapy”: 

Comment 6: Well, this would become believable if at least concrete data from some patients are described and compared to.

The reviewer is right, and we appreciate her/his input on this important point. Although this is a wishful thought for all scientists working in the field, the work performed and presented in this manuscript does not support the statement. Thus, we have omitted this statement, and the abstract has been revised as follows:

Abstract: Background/Objectives: Cancer is a dynamic and complex disease that remains largely untreated despite major advances in oncology and treatment. In this context, we aimed here to investigate optimal control techniques in the management of tumour growth inhibition, with a particular focus on cancer chemotherapy treatment strategies. Methods: Using both linear autoregressive with exogenous inputs (ARX) and advanced nonlinear tumour growth inhibition (TGI) modelling approaches, we investigated various single agent treatment protocols, including continuous, periodic and intermittent chemotherapy schedules. By integrating advanced mathematical modelling with optimal control theory and methods, namely the Linear Quadratic Regulator (LQR) and the "pseudolinear" state-space equivalent representation and suboptimal control of a non-linear dynamic sys-tem known as the State-Dependent Riccati Equation (SDRE) approach, this work explores and evaluates successfully, more effective chemotherapy treatment strategies at a computer simulation level, using real preclinical data which increases the expectation to be applied in the clinical practice of oncology. Results: The integration of these methods provides in-sights into how different drug administration schedules may affect tumour response at the preclinical level. This work uses mathematical modelling to evaluate the efficacy of various periodic and intermittent chemotherapy treatment strategies, with a focus on optimising drug doses while minimising potential side effects of chemotherapy due to the administration of less effective chemotherapeutic doses. Conclusions: The treatment scenarios tested in this study could effectively stop tumour growth or even lead to tumour regression to a negligible or near-zero size. This approach highlights the importance of computational tools for more effective treatment strategies in chemotherapy and offers a promising direction for future research and more efficient clinical applications in oncology as part of a more individualized approach.

Round 2

Reviewer 1 Report

Comments and Suggestions for Authors

This is a better version but I think it can be further improved. The main problem for the readers may be the interpretation of the models variables (x's in the first and z's in the second) used as they are newer explained. Authors provided information, that they use Simeoni models, but at least 2-3 sentenced explanation of the variables and parameters will significantly improve the paper readibility. It will allow the readers to quickly understand the models used without the necessity of reaching to the Simeoni paper. The short information what with the schedules with variable time steps between doses will also improve the paper. I will notice the above as the minor revision.

Author Response

We thank the reviewer for her/his constructive comments, which have indeed helped us to improve our manuscript and make it clearer and easier for readers to follow.

Comment

This is a better version but I think it can be further improved. The main problem for the readers may be the interpretation of the models variables (x's in the first and z's in the second) used as they are newer explained. Authors provided information, that they use Simeoni models, but at least 2-3 sentenced explanation of the variables and parameters will significantly improve the paper readibility. It will allow the readers to quickly understand the models used without the necessity of reaching to the Simeoni paper. The short information what with the schedules with variable time steps between doses will also improve the paper. I will notice the above as the minor revision.

To address reviewer’s concerns the manuscript has been revised as follows (changes are shown in red in the text)

Results

Page 6 lines 253-265:

…where  is the output of the ARX model, i.e., the tumor’s weight prediction at the discrete time , , are the past tumor’s weight observations and , are the exogenous inputs, i.e., the anticancer agent dose at each discrete time , with . The order of the ARXmodel is defined by the set of parameters. These two parameters depict the number of lags considered for the output and input historical data, respectively. Parameters  and  are weights associated with each previous observation and exogenous input, respectively. In the context of this study, represents the tumor’s weight observations, reflecting the size and growth of the tumor over time. On the other hand,  corresponds to the dosages of chemotherapy administered over time, providing a quantitative measure of the treatment intensity.

The estimated coefficients of the linear discrete time ARX (3,3) TGI model used in this study are given bellow [36,49]:

Page 7 lines 275-277

The state variables  are a set of internal variables in the linear state space representation (16) without explicit physical meaning, however determined to capture mathematically the tumor’s (output) dynamic response to chemotherapy (input).

and lines 281,282

… (i.e., the tumor’s weight at each discrete time ) and the corresponding input  (i.e., the anticancer agent dose at each discrete time ).

Page 10 line 358

…three-compartment transit TGI…

Page 11, lines 361-377

In the above non-linear model, the state variables  represent key components of the tumor's dynamic response to chemotherapy over time. Specifically:

  • : represents the proliferating portion of tumor cells, which actively contribute to tumor’s growth. This state reflects the cells that remain unaffected by the drug's cytotoxic effects at time .
  • : represent the tumor’s cells in progressive stages of damage due to the chemotherapy drug. These compartments capture the delayed effect of the drug, where damaged cells are passing through increasingly severe states of damage, before eventually dying.

In the above equations,  ( 1/day) is a first-order rate constant of tumor’s growth transit,  ( ml/ng ∙ day) measure of anticancer drug potency,  ( 1/day) and  ( g/day) are the first and zero-order rate constants of tumor’s growth (i.e., they characterize the rate of exponential and linear growth), respectively. The  is the chemotherapy drug dose in mg/kg (and not the drug’s concentration, as it is the case in the initially proposed by Simeoni et al.`s TGI model [37]), thus it is the control input at each time instance  while  is the corresponding  tumor’s weight, i.e. the non-linear model’s (18) output.

Page 15, lines 488-495

To this direction, optimal drug dosages determination in the case of intermittent chemotherapy schedules is also valuable to be explored, thus they are proposed and their efficiency is investigated in the present work. To be more precise, optimal dosages for two cases of different intermittent treatment schedules were investigated. In both cases chemotherapy was administered every three days for five times, i.e., q3dx5. To mitigate toxicity and drug resistance, the treatment is then paused for a period of  (Case B.6) and 10 days (Case B.7), respectively. Following the treatment pause (i.e. chemotherapy’s “holiday”),

Page 16, lines 509-511

Moreover, this total amount of drug reduction has also resulted in faster tumor eradication potentially reducing the chemotherapy’s toxicity.

Discussion

Page 17, lines 542-549

In both cases, periodic and intermittent optimized treatment schedules were extensively explored and compared for the first time. To mitigate excessive toxicity, strict limits were set on the doses of drugs suggested by the optimal controllers. In all the scenarios studied, including continuous, periodic and intermittent administration of the drug with treatment pauses, the size of the tumor was reduced. It is shown that the introduction of intermittent treatment schedules could serve as a potential alternative, with the aim of minimising toxicity while eradicating efficiently the tumor and thus improving the patient's overall quality of life during the treatment period.

Reviewer 2 Report

Comments and Suggestions for Authors

The revised manuscript has been considerably improved. Hence, I recommend this paper for publication in this reputable journal. 

Author Response

The revised manuscript has been considerably improved. Hence, I recommend this paper for publication in this reputable journal. 

We would like to thank the reviewer for his/her time and effort. This has helped us to improve our manuscript considerably.